# Effects of Particle Size of Curcumin Solid Dispersions on Bioavailability and Anti-Inflammatory Activities

**DOI:** 10.3390/antiox12030724

**Published:** 2023-03-15

**Authors:** Chihiro Kato, Mayuko Itaya-Takahashi, Taiki Miyazawa, Junya Ito, Isabella Supardi Parida, Hiroki Yamada, Akari Abe, Mika Shibata, Keita Someya, Kiyotaka Nakagawa

**Affiliations:** 1Laboratory of Food Function Analysis, Graduate School of Agricultural Science, Tohoku University, 468-1 Aramaki Aza-Aoba, Aoba-ku, Sendai 980-8572, Japan; 2New Industry Creation Hatchery Center (NICHe), Tohoku University, 6-6-10 Aramaki Aza-Aoba, Aoba-ku, Sendai 980-8579, Japan; 3Yokohama Oils and Fats Industry Co., Ltd., 380-7 Horiyamashita, Hadano 259-1304, Japan

**Keywords:** anti-inflammatory, bioavailability, cellular uptake, curcumin, curcumin glucuronide, intravenous administration, metabolism, nanoparticles, oral administration, particle size

## Abstract

The delivery of curcumin (CUR) using the solid dispersion system (CUR solid dispersions; C-SDs) has been shown to improve CUR bioavailability. However, it is unclear how different particle sizes of C-SDs affect the bioavailability and biological activities of CUR. Hence, we prepared C-SDs in different sizes using food-grade excipients and evaluated their bioavailability and biological activities. By pulverizing large particle sizes of C-SDs using zirconia beads, we successfully prepared C-SDs I-IV (particle size: (I) 120, (II) 447, (III) 987, (IV) 1910 nm). When administrated orally in rats, the bioavailability of CUR was increased with decreasing C-SDs size, most likely by improving its solubility in micelles. When administrated intravenously in rats, blood concentrations of CUR were increased with increasing particle size, suggesting that larger C-SDs presumably control the metabolic conversion of CUR. In RAW264 cells, more CUR was taken up by cells as their sizes reduced, and the more potent their anti-inflammatory activities were, suggesting that smaller C-SDs were taken up through a number of cellular uptake pathways. Altogether, the present study showed an evident effect of C-SDs size on their bioavailability and anti-inflammatory activities—information that serves as a basis for improving the functionality of CUR.

## 1. Introduction

Curcumin (CUR) is one of the principal lipophilic polyphenols in turmeric (*Curcuma longa* L.). While it has various physiological benefits, including anti-inflammatory, antioxidant, and lipid-lowering activities [1,2], these functions are often limited due to the low bioavailability of CUR [3,4]. Efforts have been made to improve the bioavailability of CUR [5]; among these, delivery strategies using solid dispersion techniques (i.e., CUR solid dispersions; C-SDs [6,7,8]) have gained increasing attention due to their promising efficiency.

In C-SDs, CUR is encapsulated in micro/nanoscale vehicles and dispersed in water with the aid of surfactants. Compared to bare CUR, this system improves CUR bioavailability, most likely by enhancing its solubility in bile salt micelles. In fact, our previous findings demonstrated that the delivery of CUR using poly-(lactic-co-glycolic acid) (PLGA)-based nanoparticle dispersions improved its oral bioavailability in rats by about 10-fold [9]. In another study, the delivery of CUR in solid dispersions consisting of solid lipid or D-α-tocopheryl polyethylene glycol 1000 succinate improved the oral bioavailability of CUR in rats by about 9.5-fold and about 65-fold, respectively [10,11]. Even when administered via the intravenous route, polymeric particle-based C-SDs showed beneficial curative properties in the mouse model of acute inflammation [12]. Despite the compelling results, it is important to note that these previous reports used excipients that are tailored for pharmaceutical products, aiming for disease treatments. In recent years, however, measures to prevent diseases through the consumption of functional foods have gained increasing interest, and, thus, we should consider using food-grade excipients in formulating the C-SDs to facilitate their applications in food products.

In general, the particle sizes of the encapsulated lipophilic compounds affect their biological activities when administered via oral and intravenous routes [13,14]. For food-grade C-SDs, it was unclear how their particle sizes may affect the bioavailability of CUR. Hence, it is vital that we understand the effect of particle sizes on the bioavailability of food-grade C-SDs in the first place. Some studies have provided salient examples of the significance of compound particle sizes on their biological activities. For instance, previous studies have demonstrated the increasing bioaccessibility of some lipophilic compounds (i.e., β-carotene, coenzyme Q10, and raloxifene) after oral administration, with decreasing particle sizes in dispersions comprised of bile acids or surfactants [15,16,17]. Furthermore, even for intravenous administration, optimizing particle size is known to be important (e.g., for blood profiles) [18]. Considering the above points, we aimed to formulate C-SDs using food-grade excipients with a strictly controlled particle size, which is expected to optimally enhance the bioavailability of CUR for a wide range of applications, including disease prevention and treatment.

To achieve our objective, we attempted to prepare four C-SDs with different particle sizes ((I) 120, (II) 447, (III) 987, (IV) 1910 nm) using food-grade excipients. Once we successfully prepared the C-SDs, we determined how their particle sizes affect their functional potencies by investigating: (1) their oral bioavailability based on the plasma level of CUR and its metabolites (e.g., curcumin glucuronide (CURG)) post-oral administration in rats, (2) their blood profile after intravenous administration in rats, and (3) their cellular uptake and anti-inflammatory activity through cell experiments (Figure 1). The findings from the present study will hopefully be useful in efforts to improve the functionalities of bioactive food compounds such as CUR and formulate functional food products.

## 2. Materials and Methods

### 2.1. Reagents

Lipopolysaccharide (LPS), N-(1-naphthyl) ethylenediamine, sodium nitrite, and sulfanilamide were purchased from Wako Pure Chemical Industries, Ltd. (Osaka, Japan). All other chemicals and reagents used in the study were of analytical grade or higher.

### 2.2. Preparation of C-SDs

We started by determining suitable materials for preparing C-SDs I-IV under optimized conditions. To prepare the excipient emulsion, 5.0 g of glycerin, 3.5 g of decaglycerol monoester, 2.0 g of lysolecithin, and 0.2 g of sodium chloride were dispersed in 83.8 mL of water then heated at 70 °C for 30 min. Into the emulsion, 5.5 g of the CUR powder (Vidya Japan K. K, Tokyo, Japan) was gradually added over 30 min with constant stirring at 6000 rpm using a homomixer (Homomixer mark II; PRIMIX Corp., Hyogo, Japan), and 1.3 mL of ethanol was mixed at room temperature to obtain C-SDs IV. To obtain C-SDs III, 80 mL of C-SDs IV was transferred to a planetary mill (Pulverisette 6; Fritsch Japan Co., Ltd., Kanagawa, Japan) and pulverized with zirconia beads (mean diameter: 0.3 mm) at 350 rpm for 2.5 min. C-SDs II was obtained by further pulverizing 60 mL of C-SDs III in a planetary mill for 15 min, and C-SDs I was obtained by pulverizing 40 mL of C-SDs II in a planetary mill using smaller zirconia beads (average diameter: 0.03 mm) at 600 rpm for 180 min. C-SDs I-IV were stored at room temperature and shielded from light until use in experiments. The range of particle size was determined based on a previous study [19].

### 2.3. Characterization of C-SDs I-IV

To determine the particle size and zeta potential of C-SDs I-IV, 20 µL of C-SDs I-IV were diluted with water to appropriate concentrations and analyzed using dynamic light scattering and laser Doppler anemometry (ELS-Z; Otsuka Electronics Co., Ltd., Osaka, Japan).

To observe the shape of C-SDs I-IV, 20 µL of C-SDs I-IV were diluted 1000-fold with water and photographed using a transmission electron microscope (TEM, H-7650 ZeroA; Hitachi, Ltd., Tokyo, Japan).

To measure the CUR concentration in C-SDs I-IV, 100 µL of C-SDs I-IV were dissolved in 30 mL of methanol and passed through a filter to remove insoluble components (GL Chromatodisc, 0.45 µm; GL Science, Tokyo, Japan). One hundred microliters of the solution were diluted with 900 µL of methanol, and a 10 µL aliquot was subjected to analysis using high-performance liquid chromatography with ultraviolet detection (HPLC-UV; JASCO, Tokyo, Japan). Chromatographic separation was performed using a ProC18 column (4.6 × 150 mm, 5 µm; YMC Co., Ltd., Kyoto, Japan) with a binary gradient consisting of solvent A (0.05% formic acid) and solvent B (acetonitrile). The gradient profile was as follows: 0–17 min, 30–50% B linear; 17–22 min, 50–100% B linear; 22–32 min, 100% B. The flow rate was adjusted to 1.0 mL/min, and the column temperature was maintained at 40 °C. CUR was detected at 420 nm, and its concentration was determined using the standard curve of CUR (Nagara Science Co., Ltd. (Gifu, Japan)).

### 2.4. Bioavailability of C-SDs after Oral Administration in Rats

Seven-week-old male Sprague Dawley (SD) rats weighing 220–240 g were purchased from CLEA Japan Inc., (Tokyo, Japan). The rats were group-housed (two rats per cage) and placed in the animal experimental room with a 12 h light/dark cycle and controlled temperature of 24 °C. Prior to the experiment, rats were acclimated for one week with ad libitum access to water and commercial rodent chow (CE-2; CLEA Japan Inc., Tokyo, Japan).

For oral administration, acclimated rats were fasted overnight and randomly divided into four groups (C-SDs I-IV, n = 4). Each group received an oral administration of C-SDs I-IV by a gastric tube equivalent to 100 mg CUR/kg BW (380–504 µL depending on BW). The dose of C-SDs was determined based on a previous study [9] and the preliminary experiment. Blood was collected from the tail vein in heparinized tubes at 0, 1, 3, 6, 12, and 24 h following oral administration and centrifuged at 3600 rpm for 5 min at 4 °C to obtain plasma (MX-307; TOMY SEIKO Co., Ltd., Tokyo, Japan). Plasma samples (100 µL or less) were then stored at −80 °C until analysis.

One hundred microliters of plasma was mixed with 500 µL of acetonitrile, vortexed, and centrifuged at 7400 rpm for 5 min at 4 °C (MX-307; TOMY SEIKO Co., Ltd., Tokyo, Japan). The supernatant was collected, 400 µL of water was added, and it was filtered to remove insoluble components (GL Chromatodisc, 0.45 µm; GL Science, Tokyo, Japan). The same ratio of acetonitrile and water was used for plasma volumes less than 100 µL. Ten microliters of the filtrate were loaded into a high-performance liquid chromatography-tandem mass spectrometry machine (HPLC-MS/MS, 4000 QTRAP; Sciex, Redwood City, CA, USA). Chromatographic separation was performed using an Xbridge™ C18 column (2.1 × 150 mm, 3.5 µm; Waters, Milford, MA, USA) with a binary gradient consisting of solvent A (0.05% formic acid) and solvent B (acetonitrile). The gradient profile was as follows: 0–20 min, 15–100% B linear. The flow rate was adjusted to 0.2 mL/min, and the column temperature was maintained at 40 °C. CUR and CURG were detected using electrospray ionization (ESI) MS/MS in negative ionization mode using the following multiple-reaction monitoring (MRM) transitions: CUR, m/z 367 > 217 (collision energy (CE), −18 V; declustering potential (DP), −60 V); CURG, m/z 543 > 134 (CE, −72 V; DP, −85 V). The following MS setting was used for the optimal detection of CUR and CURG: turbo gas temperature, 500 °C; spray voltage, −4500 V; ion source gas 1, 60 psi; ion source gas 2, 50 psi; curtain gas, 30 psi; collision gas, 3.0 V. Concentrations of CUR and CURG were determined according to the standard curves of CUR (Wako Pure Chemical Industries, Ltd., Osaka, Japan) and CURG (Therabiopharma, Ltd., Kanagawa, Japan), respectively.

### 2.5. Blood Profile of C-SDs after Intravenous Administration in Rats

Preliminary feeding was carried out using the same protocols as mentioned in Section 2.4. For intravenous administration, acclimated rats were randomly divided into three groups (C-SDs I-III, n = 4). Each group received intravenous doses of C-SDs I-III (diluted 5-fold with saline) equivalent to 5 mg CUR/kg BW (104–138 µL depending on BW) via the tail vein. The dose of C-SDs was determined based on the preliminary experiment. From the viewpoint of animal ethics, C-SDs IV, which easily aggregates, was not administered intravenously. Two hours after the intravenous administration of C-SDs I-III, rats were sacrificed, and blood was drawn by cardiac puncture using a heparinized syringe, after which the rats were perfused with saline. Plasma was obtained from blood using the same method as previously mentioned in Section 2.4. To measure the concentrations of CUR and CURG in plasma, 100 µL of plasma underwent extraction and quantification protocols as previously mentioned in Section 2.4.

All animal studies were conducted in accordance with the Committee on the Ethics of Animal Experiments and carried out in accordance with the Animal Experiment Guidelines of Tohoku University (Sendai, Japan). The permit number for this animal experiment is 2021–AgA–011.

### 2.6. Cellular Uptake and Anti-Inflammatory Activity of C-SDs

Mouse leukemic monocyte macrophage (RAW264) cells were obtained from Japan Food Research Laboratories (Tokyo, Japan) and maintained in Dulbecco’s modified Eagle medium (DMEM D-6429; 4500 mg/L glucose, high pyruvate; Sigma-Aldrich, St. Louis, MO, USA) supplemented with 10% fetal bovine serum (FBS) and antibiotics (100 U/mL penicillin and 100 µg/mL streptomycin) in 100 mm plastic Petri dishes at 37 °C and 5% CO_2_. Cell lines were determined based on a previous study [20].

To measure the cellular uptake of CUR after exposure to C-SDs I-IV, RAW264 cells (1.0 × 10⁴ cells) were seeded in 35 mm glass bottom dishes (Matsunami Glass Industries, Osaka, Japan) and grown in 3 mL of medium. After overnight incubation, the medium was replaced with 3 mL of medium containing C-SDs I-IV (final concentration of 10 µM CUR). After another round of incubation for 15 min, cells were washed with phosphate-buffered saline, and CUR in C-SD I-IV-treated cells was observed using confocal laser microscopy (LSM 710; Carl-Zeiss, Baden-Wurttemberg, German). An excitation wavelength of 488 nm and emission wavelength of 519 nm were used to detect the fluorescent of CUR itself.

To measure nitric oxide (NO) production, RAW264 cells (1.0 × 10⁴ cells/well, n = 6) were seeded in a 96-well plate in 100 µL medium. After a 24 h incubation, 20 µL of medium containing LPS and C-SDs I-IV was added into each well. The final concentration of LPS in the medium reached 100 ng/mL, and the addition of C-SDs I-IV resulted in the final concentration of CUR being 0–15 µM. After further incubation for 24 hours, 80 µL of the supernatant was transferred to another 96-well plate and NO levels were measured through a colorimetric assay as described in a previous study [20].

To measure the expression of genes related to inflammatory response, we performed the reverse transcription-quantitative polymerase chain reaction (RT-qPCR) and enzyme-linked immunosorbent assay (ELISA). For RT-qPCR analysis, RAW264 cells were seeded in 60 mm dishes in a 5 mL medium at a density of 1.0 × 10⁶ cells (n = 3). For ELISA, RAW264 cells were seeded in 60 mm dishes in a 5 mL medium at a density of 1.0 × 10⁶ cells (for IL-1β, n = 3) and 2.5 × 10⁵ cells (for IL-6 and MCP-1, n = 3). After incubation for 24 h, the medium was replaced with 5 mL of medium containing LPS (100 ng/mL) and C-SDs I-IV to reach a final concentration equivalent to 10 µM of CUR. After another 24 h of incubation, cells were harvested to prepare RNA extract for RT-qPCR assay, while the mediums were collected for ELISA. Total RNA was extracted from cell lysates, and cDNA was synthesized according to methods previously described [21]. The gene expressions of pro-inflammatory markers (*Il-1β*, *Cox-2*, *Il-6*, *Mcp-1*, *Inos*, *Tnf-α*, *Nf-κb*, *Tgf-β*, *Ho-1*, *Cat,* and *Nrf-2*) were evaluated through RT-qPCR as described in a previous study [21], and the sequences of primers used in the present study are listed in Table 1. For the detection of *Il-1β*, *Mcp-1*, *Inos*, *Tnf-α*, *Cox-2*, *Ho-1*, *Tgf-β,* and *Nrf-2*, samples were subjected to the following amplification condition using an RT-qPCR machine (C1000™ Thermal Cycler; BIO-RAD, Hercules, CA, USA): 30 s at 95 °C, 5 s at 95 °C (40 cycles), and 30 s at 60 °C. The annealing temperature was set at 63.3 °C for *Cat* and 64.5 °C for *Il-6* and *Nf-κb*, while the rest of the parameters remained similar. The relative expression of each target gene was determined using the 2^−ΔΔCt^ with housekeeping gene *Gapdh* as the reference gene. The inflammatory cytokines (IL-6, MCP-1, and IL-1β) in the collected mediums were measured using ELISA kits (ELISA MAX Deluxe Set; BioLegend ltd., San Diego, CA, USA) as described in a previous study [20]. The concentration of CUR was determined based on preliminary experiments.

### 2.7. Statistical Analysis

Data are expressed as mean ± standard deviation (SD) or mean ± standard error (SE). Differences between multiple groups were determined using one-way analysis of variance (ANOVA) and Tukey’s posthoc test with *p* < 0.05 considered statistically significant. For comparison between two treatment groups, ANOVA and Dunnett’s posthoc test was performed, with *p* < 0.05 and *p* < 0.01 considered statistically significant. All statistical analyses were performed using the statistics software (EZR; Jichi Medical University Saitama Medical Center, Saitama, Japan) [22].

## 3. Results and Discussion

### 3.1. Characterization of C-SDs

Previous reports have shown an increase in the biological activities of CUR delivered by various nanoparticle systems, either through oral or intravenous routes [5,23]. However, most available reports on C-SDs used pharmaceutical excipients in one particular size. Hence, in the present study, we prepared C-SDs with varying particle sizes at the submicron level using food-grade excipients; the physicochemical properties of C-SDs were characterized, and their bioavailability and efficacy were evaluated. 

To prepare the C-SDs, we incorporated lecithin and decaglycerol monoester in C-SDs matrices as they are widely used as food-grade excipients and considered safe for oral administration [24,25]. The formation of C-SDs or any encapsulation process, in general, consists of two main steps: the emulsification of polymer substances containing encapsulated compounds using high-pressure homogenization [26] and the hardening of the nanoparticle surface through solvent evaporation [8,27,28]. However, these methods were not suitable for producing C-SDs in multiple particle sizes, especially using the same excipients, as CUR tends to form aggregates. To overcome this problem, we instead pulverized the homogenized dispersion using zirconia beads; by adjusting the size of beads and grinding time, we generated C-SDs I-IV with particle sizes ranging from submicron to micron using the same food-grade excipients. Using dynamic light scattering, we determined the particle diameter of C-SDs as follows: C-SDs I, 122 nm; C-SDs II, 448 nm; C-SDs III, 991 nm; and C-SDs IV, 1910 nm (Table 2). These were in line with their TEM images (Figure 2). C-SDs I-IV had negative zeta potentials, suggesting that they are properly encapsulated in lecithin and decaglycerol monoester [29,30]. Using the HPLC-UV, we found the following concentration of CUR in C-SDs: 176.7 mM in C-SDs I; 202.5 mM in C-SDs II; 163.6 mM in C-SDs III; and 172.0 mM in C-SDs IV. This showed that almost all CUR was dispersed in water without undergoing oxidation or degradation. In addition to CUR, the HPLC-UV chromatogram of C-SDs I-IV showed peaks of bisdemethoxycurcumin (BDMC) and demethoxycurcumin (DMC) (Appendix A) [4], which are the analogs of CUR commonly present in turmeric products along with CUR itself. Overall, CUR serves as the major active constituent in C-SDs, with BDMC and DMC present at 1/40 and 1/8 of the amount of CUR, respectively. This level of CUR is sufficient for evaluating the biological activities of C-SDs in the following experiments. C-SDs I-IV were well-dispersed through sonication before the experiment.

### 3.2. Oral Bioavailability of C-SDs

To examine how the particle sizes of C-SDs affect the oral bioavailability of CUR, we measured the changes in plasma levels of CUR and its metabolites in rats within 24 h of oral administration of C-SDs I-IV (=100 mg/kg BW of CUR per administration). Initially, CUR and its metabolites were not detected in rat plasma prior to C-SDs I-IV administration. Irrespective of the particle sizes of C-SDs, CURG was detected as the predominant constituent in plasma, followed by CUR, within 24 h post-loading (Figure 3). Other metabolites were also detected, although to a lesser extent (Appendix A), and the highest area under the plasma concentration–time curve (AUC) level of CURG was exhibited by the C-SDs I group (Figure 4). Thus, these results suggest that reducing the particle size of C-SDs does improve the bioavailability of CUR, but the differences in particle sizes may not affect CUR metabolism, which commonly occurs in the intestine, liver, and kidney [31]. The relationship between particle sizes and oral bioavailability has been reported for compounds other than CUR. For example, β-carotene in smaller particle sizes (120 nm, 190 nm, 14,000 nm) showed better solubility in micelles [15]. Furthermore, an ex vivo study reported how bovine serum albumin (BSA) in small particles (100 nm) penetrated rat intestinal submucosa at a higher rate compared to large particles (500 nm, 1 μm, and 10 μm) [32]. Therefore, we infer that C-SDs I increased the oral bioavailability of CUR by improving its solubility in bile acid micelles (as described in the Introduction) and/or increasing its penetration into the intestinal submucosa; future studies should focus on elucidating the effect of C-SDs size on these aspects. Additionally, the AUC of CURG in the C-SDs I administered group was consistent with our previous study using C-SDs with PLGA [9], a pharmaceutical excipient [33]. All things considered, our data showed that the small particle size of C-SDs prepared from food-grade excipients improves the bioavailability of CUR to the same extent as the ones made with pharmaceutical excipients.

### 3.3. Blood Profile of C-SDs after Intravenous Administration

C-SDs are often used not only for oral administration but also for intravenous use [12]; thus, we evaluated the effect of particle sizes on the CUR blood profile following the intravenous administration of C-SDs I-III (=5 mg/kg BW of CUR per administration). Similar to the results from oral administration (Section 3.2), CUR and CURG were not detected in rat plasma prior to intravenous administration. At two hours post-loading, we detected CURG as the predominant constituent in plasma, followed by CUR (Figure 5). Other metabolites were also detected, though in lesser amounts. This suggests that even when administered intravenously, a relatively large amount of CUR is metabolized into CURG, an event that is likely to take place in the liver and kidney instead of the intestinal tract [31]. It is interesting to note that while the blood concentration of CURG remained unaffected, the concentration of CUR increased with increasing particle size (Figure 5), thereby suggesting that we can control the metabolic conversion of CUR to CURG by adjusting the particle size of C-SDs. Thus, by increasing the particle size of C-SDs and inhibiting the metabolic conversion of CUR to CURG, we are more likely to increase the in vivo activities of CUR as it is generally more potent compared to CURG [34]. It should be pointed out that a considerable accumulation of CUR was found in the lungs and liver of rats given the larger size C-SDs (II and III) (Appendix A), so future evaluation should also look out for any potential side effects due to this deposition. While some questions have yet to be answered, our results suggest that the intravenous administration of C-SDs with optimized particle size affects the metabolism rate of CUR into CURG, a phenomenon that is not achievable through oral administration.

### 3.4. Cellular Uptake and Anti-Inflammatory Activities of C-SDs

From Section 3.2 and Section 3.3, it appears that the particle size of C-SDs affects CUR bioavailability when administered via the oral route, and presumably affects its metabolism when given via the intravenous route. Previous studies have reported C-SDs interaction with immune cells and anti-inflammatory activities after intravenous administration [12]; hence the present study assessed the effect of C-SDs particle sizes on their cellular uptake and anti-inflammatory activities in the RAW264 macrophage cell line. First, RAW264 cells were incubated in a medium containing C-SDs I-IV, and CUR fluorescence was detected through confocal laser microscopy. The smaller the particle sizes, the higher the fluorescence intensity that is considered to be derived from CUR (Figure 6). Then, the effect of C-SDs particle sizes on their anti-inflammatory activities was assessed based on NO production, gene expression, and secretion of inflammatory cytokines. We found that NO production was inhibited in a dose-dependent manner (5–15 µM) in C-SDs I-IV-treated cells, with smaller particle sizes leading to greater inhibition (Figure 7). There were no significant differences in the cell viability between groups, confirming that cell death did not affect NO production (Appendix A). Along with the aforementioned changes, the expression levels of some pro-inflammatory genes (*Il-1β*, *Cox-2*, and *Il-6*) were reduced by C-SDs treatment, particularly in the C-SDs I and II-treated groups (Figure 8A–C). In addition, the secretion levels of inflammatory cytokines (IL-1β, IL-6, and MCP-1) were markedly suppressed as the particle size decreased (Figure 9). Altogether, our data showed that the smaller the particle sizes, the more efficiently CUR was transferred into the cells and the more potent the anti-inflammatory activities. Small-sized particles can be taken up by cells via multiple pathways, such as clathrin-mediated endocytosis, caveolae-mediated endocytosis (for nanoscale particles), and phagocytosis (nano- to micro-scale particles) [35]. The efficiency of these pathways depends on the particle sizes of the transported particles, which likely explains why smaller C-SDs were taken into cells at a higher rate. To identify the mechanisms involved in the cellular uptake of CUR encased in C-SDs, it may be necessary for future studies to assess C-SDs uptake in the presence of the inhibitors of each pathway. In addition, our previous data showed that piperine increases the cellular uptake of CUR by competitively inhibiting the binding of CUR to BSA [20], and, therefore, we are interested in creating C-SDs containing CUR and piperine as this is likely to further increase its cellular uptake.

## 4. Conclusions

In the present study, C-SDs with different particle sizes were successfully prepared using the same food-grade excipients by pulverizing large-sized C-SDs using zirconia beads. When administered orally, our small-sized C-SDs were able to improve CUR bioavailability to the same extent as the ones made with pharmaceutical excipients (AUC of CURG, 21,364.8 ± 3048.8 nM-hours). When administered intravenously, larger-sized C-SDs appeared to inhibit the metabolic conversion of CUR to CURG about 1.6 times more than smaller sized C-SDs. Cell experiments showed higher cellular uptake of CUR when delivered in smaller-sized C-SDs, and accordingly, their anti-inflammatory activities were more potent. From these findings, it is inferred that we can modify the bioavailability and biological activities of CUR by adjusting the particle sizes of food-grade C-SDs according to the intended purposes. Altogether, this study provides a basis for improving the functionality of CUR and developing new food products containing CUR. For this purpose, it would be essential for future research to conduct short- and long-term pharmacokinetic studies in humans and identify any potential side effects from C-SDs intake (e.g., on bodyweight change and psychokinetic behaviors).

## Figures and Tables

**Figure 1 antioxidants-12-00724-f001:**
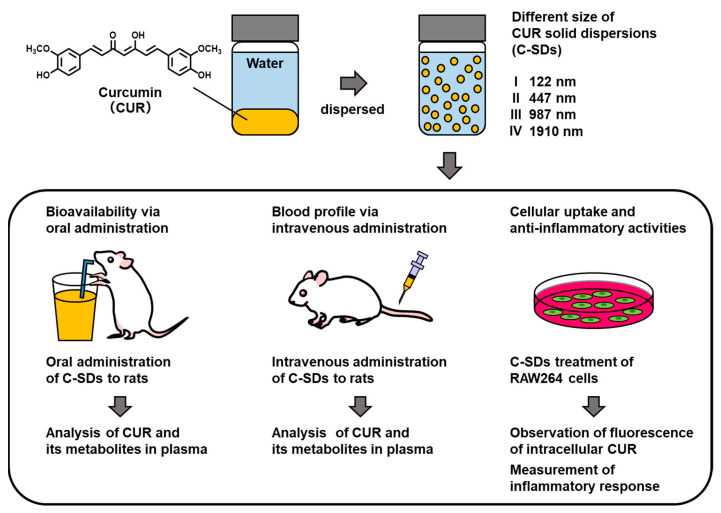
The experimental scheme of the present study.

**Figure 2 antioxidants-12-00724-f002:**
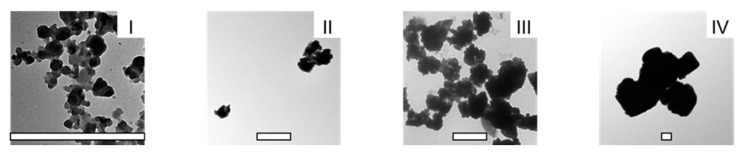
TEM images of C-SDs I-IV. Shape of C-SDs I-IV was evaluated through TEM (C-SDs I:20,000; C-SDs II:6,000; C-SDs II:5,000; C-SDs IV:3,000 magnification each). Scale bar in each image is 1 µm. C-SDs, curcumin solid dispersions; TEM, transmission electron microscope; I, C-SDs I; II, C-SDs II; III, C-SDs III; IV, C-SDs IV.

**Figure 3 antioxidants-12-00724-f003:**
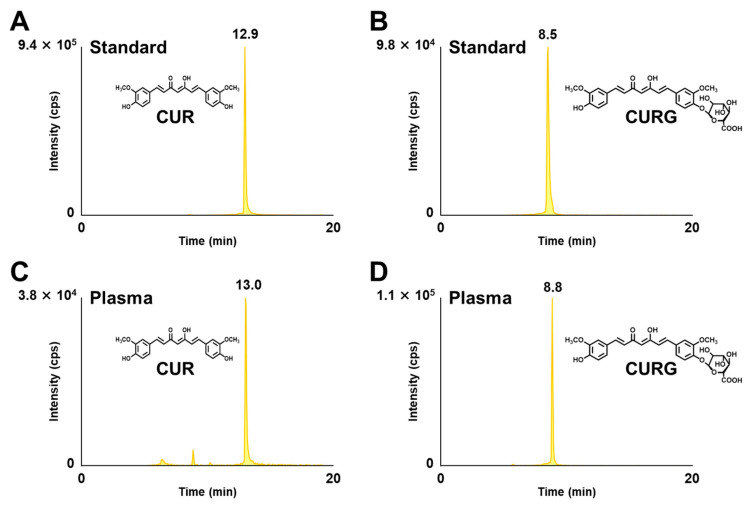
Representative HPLC-MS/MS chromatograms of CUR and CURG in plasma following oral administration. (**A**) Standard CUR (2 pmol/injection), (**B**) standard CURG (2 pmol/injection), (**C**) CUR in rat plasma at one hour following oral administration of C-SDs I, and (**D**) CURG in rat plasma at one hour following oral administration of C-SDs I. CUR, curcumin; CURG, curcumin glucuronide; C-SDs, curcumin solid dispersions.

**Figure 4 antioxidants-12-00724-f004:**
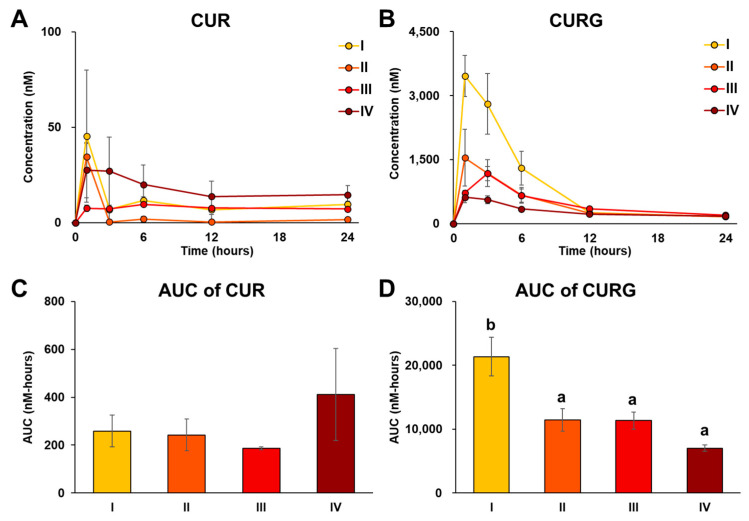
Plasma concentration–time profiles and AUC of CUR and CURG in the C-SDs I-IV oral administration groups. Plasma concentration–time profiles of (**A**) CUR and (**B**) CURG. Area under the blood concentration–time curve (AUC) was used to compare (**C**) CUR and (**D**) CURG between groups (mean ± SE, n = 4, ^a,b^
*p* < 0.05 (Tukey’s posthoc test)). There is no statistically significant difference between groups in Figure 4C. CUR, curcumin; CURG, curcumin glucuronide; C-SDs, curcumin solid dispersions; I, C-SDs I; II, C-SDs II; III, C-SDs III; IV, C-SDs IV.

**Figure 5 antioxidants-12-00724-f005:**
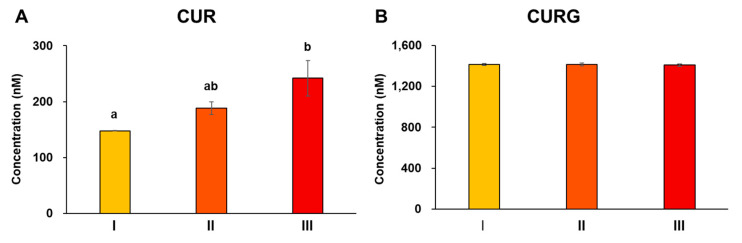
Plasma concentration of CUR and CURG in the C-SDs I-IV intravenous administration groups. Concentration of (**A**) CUR and (**B**) CURG in plasma following intravenous administration of C-SDs I-III at two hours (mean ± SE, n = 4, ^a,b^
*p* < 0.05 (Tukey’s posthoc test)). There is no statistically significant difference between groups in Figure 5B. CUR, curcumin; CURG, curcumin glucuronide; C-SDs, curcumin solid dispersions; I, C-SDs I; II, C-SDs II; III, C-SDs III.

**Figure 6 antioxidants-12-00724-f006:**
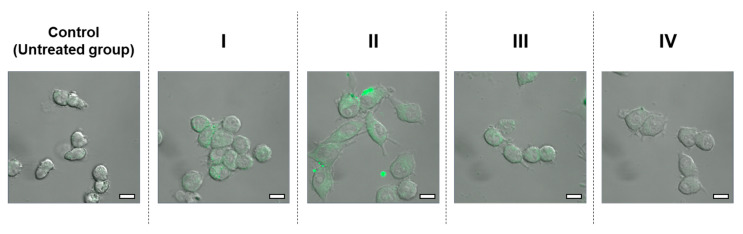
Uptake of C-SDs I–IV into RAW264 cells. Excitation wavelength: 488 nm, emission wavelength: 519 nm. C-SDs, curcumin solid dispersions; I, C-SDs I; II, C-SDs II; III, C-SDs III; IV, C-SDs IV. Scale bar in each image is 10 µm.

**Figure 7 antioxidants-12-00724-f007:**
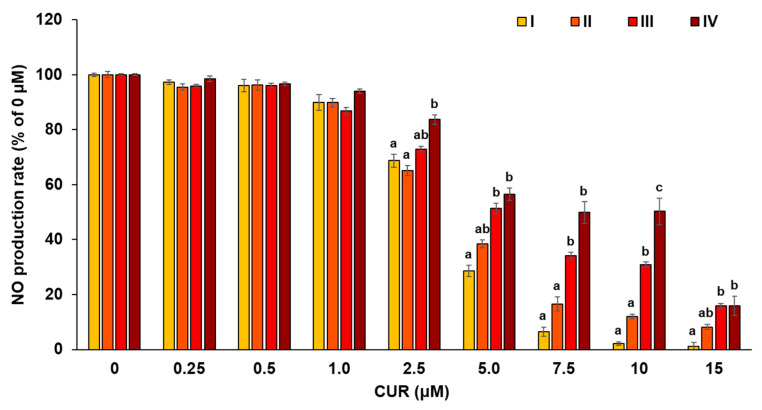
NO production in RAW264 cells. NO production and cell viability in LPS-stimulated RAW264 cells with C-SDs I-IV (^a,b,c^
*p* < 0.05 (Tukey’s posthoc test)). There is no statistically significant difference between groups in 0–1.0 µM. NO production in LPS-stimulated RAW264 cells (positive control, 0 µM) is 100%. C-SDs, curcumin solid dispersions; I, C-SDs I; II, C-SDs II; III, C-SDs III; IV, C-SDs IV.

**Figure 8 antioxidants-12-00724-f008:**
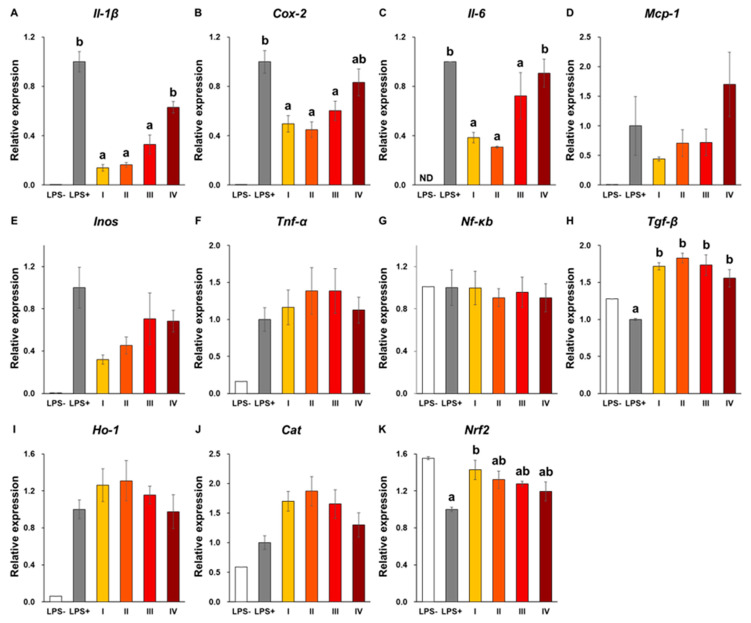
Gene expression levels in RAW264 cells. Gene expression levels ((**A**) *Il-1β*, (**B**) *Cox-2*, (**C**) *Il-6*, (**D**) *Mcp-1*, (**E**) *Inos*, (**F**) *Tnf-α*, (**G**) *Nf-κb*, (**H**) *Tgf-β*, (**I**) *Ho-1*, (**J**) *Cat*, (**K**) *Nrf2*) in LPS-stimulated RAW264 cells with C-SDs I-IV (mean ± SE, n = 3. ^a,b^
*p* < 0.05 (Tukey’s posthoc test)). There is no statistically significant difference between groups in Figure 8D–G,I,J. *Cat*, catalase; *Cox-2*, cyclooxygenase-2; C-SDs, curcumin solid dispersions; *Gapdh*, Glyceraldehyde 3-phosphate dehydrogenase; *Ho-1*, heme oxygenease 1; *Il-1β*, interleukin 1β; *Il-6*, interleukin 6; *Inos*, inducible nitric oxide synthase; *Mcp-1*, monocyte chemotactic protein 1; ND, not detected; *Nf-κb*, nuclear factor kappa b cells; *Nrf2*, nuclear factor erythroid 2 related factor; *Tgf-β*, transforming growth factor β; *Tnf-α*, tumor necrosis factor α; I, C-SDs I; II, C-SDs II; III, C-SDs III; IV, C-SDs IV.

**Figure 9 antioxidants-12-00724-f009:**
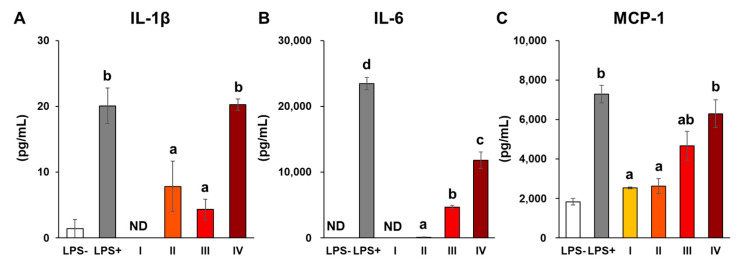
Secretion of inflammatory cytokines in RAW264 cells. Secretion of inflammatory cytokines ((**A**) IL-1β, (**B**) IL-6, (**C**) MCP-1) in LPS-stimulated RAW264 cells with C-SDs I-IV (mean ± SE, n = 3. ^a,b,c,d^
*p* < 0.05 (Tukey’s posthoc test)). C-SDs, curcumin solid dispersions; Il-1β, interleukin 1β; IL-6, interleukin 6; MCP-1, monocyte chemotactic protein 1; ND, not detected; I, C-SDs I; II, C-SDs II; III, C-SDs III; IV, C-SDs IV.

**Table 1 antioxidants-12-00724-t001:** Sequence of primers used in the present study. *Cat*, catalase; *Cox-2*, cyclooxygenase-2; *Gapdh*, Glyceraldehyde 3-phosphate dehydrogenase; *Ho-1*, heme oxygenase 1; *Il-1β*, interleukin 1β; *Il-6*, interleukin 6; *Inos*, inducible nitric oxide synthase; *Mcp-1*, monocyte chemotactic protein 1; *Nf-κb*, nuclear factor kappa b cells; *Nrf2*, nuclear factor erythroid 2 related factor; *Tgf-β*, transforming growth factor β; *Tnf-α*, tumor necrosis factor α.

Gene	Genbank ID	Forward Primer (5′-3′)	Reverse Primer (5′-3′)
*Il-1β*	NM_008361.4	TCCAGGATGAGGACATGAGCAC	GAACGTCACACACCAGCAGGTTA
*Cox-2*	NM_011198.5	TCTGGTGCCTGGTCTGATGATGT	AGTCTGCTGGTTTGGAATAGTTGCT
*Il-6*	X54542.1	ACACATGTTCTCTGGGAAATCGT	AAGTGCATCATCGTTGTTCATACA
*Mcp-1*	NM_011333.3	AGGTCCCTGTCATGCTTCTGG	CTGCTGCTGGTGATCCTCTTG
*Inos*	M87039.1	GGAATCTTGGAGCGAGTTGTGGA	GTGAGGGCTTGGCTGAGTGAG
*Tnf-α*	X02611.1	GAAAGCATGATCCGCGACGT	CGAAGTTCAGTAGACAGAAG
*Nf-κb*	XM_021152061.2	TGAAGAAGCGAGACCTGGAGCAA	GCACTGTCACCTGGAAGCAGAG
*Tgf-β*	NM_011577.2	AGACATTCGGGAAGCAGTGC	AAAGACAGCCACTCAGGCGT
*Ho-1*	NM_010442.2	AGACCGCCTTCCTGCTCAAC	ACGAAGTGACGCCATCTGTGA
*Cat*	NM_009804.2	AGCCAGAAGAGAAACCCACAGACT	AAGCCTTCCGCCTCTCCAACA
*Nrf-2*	NM_010902.5	CTTCCATTTACGGAGACCCA	ATTCACGCATAGGAGCACTG
*Gapdh*	BC023196.2	CATGTTCCAGTATGACTCCACTC	GGCCTCACCCCATTTGATGT

**Table 2 antioxidants-12-00724-t002:** Mean diameter and zeta potential of C-SDs I-IV. Mean diameter (n = 2) and zeta potential (mean ± SD, n = 7) of C-SDs I-IV were measured using dynamic light scattering and laser Doppler anemometry. C-SDs, curcumin solid dispersions.

Classification	C-SDs I	C-SDs II	C-SDs III	C-SDs IV
Mean of particle diameters (nm) (Measured value (nm))	122 (123, 121)	447 (448, 445)	987 (991, 983)	1910 (1903, 1914)
Zeta potential (mV)	−33.4 ± 1.9	−33.5 ± 3.3	−38.1 ± 3.1	−26.6 ± 7.6

## Data Availability

The datasets used and/or analyzed during the current study available from the corresponding author on reasonable request.

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
