# Peer review of "Effects of Particle Size of Curcumin Solid Dispersions on Bioavailability and Anti-Inflammatory Activities"

_antioxidants, 2023, doi:10.3390/antiox12030724_

Round 1

Reviewer 1 Report

The article "Effects of Particle Size of Curcumin Solid Dispersions on Bioavailability and Anti-inflammatory Activities" describes the effects of particles size on bioavailability for curcumin. It is a valuable study that can be published after authors address the following problems:

Motivation must be clear and authors should better explain the reasons behind chosen this system and their findings in a critical way. In introduction the hypothesis and work plan should be clearly stated.

The English language needs minor polishing for style and typos.

This work is interesting and cab be boosted further. Hence the following literature must be cited in this work as it can prove this manuscript: doi: 10.3390/pharmaceutics15020660; doi: 10.3390/pharmaceutics14051057.

The data processing is not standardized, for example lack of significant analysis in some sets.

In figure 3C results should also be interpreted statistically as in 3D. Same is true for figures 4A and 4B.

In figure 6 should be indicate if there are no statistic meaningful differences at lower concentrations. Same for all graphs in Figure 7.

The conclusion should reflect the heuristic of the study. Conclusion section must be reworked to underline the novelty and advantages of this research, with actual numbers.

Reviewer 2 Report

This manuscript describes the development of curcumin delivery systems to enhance the bioavailability of CUR for anti-inflammatory applications. This is a very interesting and well-designed study that definitely contributes to the field of addressing the limitations of curcumin’s low solubility. The synthesized C-SDs were fully characterized by several physicochemical assays. However, the standard protocol of reporting loading capacity/encapsulation efficacy/release rate is missing, something that authors must report. Furthermore, the in vitro/in vivo evaluation provides important data and the conclusion are fully supported by the results. A comment concerning any side effects like weight loss, psychokinetic behavior, etc. after the administration must also be added in the manuscript. Furthermore, authors must insert a scale bar or a magnification info in Figure 1 & 5.  

Reviewer 3 Report

Since curcumin presents with a multitude of beneficial effects on the organism, strategies to improve its bioavailability are an important aspect of ethnopharmacology. Rather than taking on a traditional approach based on nanoparticles, this study focuses on solid dispersions that could be eventually used for a direct consumption. This aspect is very interesting and promising.

The study is very complex as it incorporates an in vivo as well as an in vitro approach, and two administration routes.  In this sense, I believe the manuscript could benefit from an experimental scheme that could serve as a comprehensive review and a fast guide for the reader.  

The manuscript is well structured and presents with a comprehensive approach to evaluate the study hypothesis. Several advanced methods have been applied to reach the goals of the study, providing a complex analysis. The introduction part provides sufficient background, although several sentences on the reasons for the low CUR bioavailability could be informative. The aim is clear, and methodology is appropriate. The presentation of the results is comprehensive, supported by multiple figures and tables.

I only have a few comments:

-          How was the CUR suspension administered orally? On a vehicle or by a gastric tube?

-          Based on what reasons was the size of the particles or CUR concentration selected? Also, why did the authors decide to use leukemic monocyte macrophages as the cellular model?

-          Please, increase the size of Figure 7, the graphs are a bit difficult to read.

-          Please, briefly discuss any limitations of the study.
